# TACKLING MOLECULE ASSEMBLY WITH GRAPH DIFFUSION

## ABSTRACT

A common starting point for drug design is to find small chemical groups or "fragments" that form interactions with distinct subregions in a protein binding pocket. However, once suitable fragments are identified, assembling these fragments into a high affinity drug with desirable pharmacological properties is difficult. This "molecule assembly" task is particularly challenging because, initially, fragment positions are known only approximately, and the combinatorial space of potential connectivities is extremely large. Even if the individual fragments form favorable interactions with regions of the pocket, a poor assembly of these fragments can drastically compromise the molecule's druglikeness and hinder its binding affinity. In this paper, we present EdGr, a new graph diffusion framework tailored for the molecule assembly task. EdGr can handle both fragments and atoms, and predicted candidate edge likelihoods influence node position updates during the diffusion denoising process, allowing connectivity cues to guide spatial movements, and vice versa. EdGr substantially outperforms previous methods on the molecule assembly task and stays robust even as confidence in fragment placement decreases.

## 1 INTRODUCTION

Most small molecule drugs work by binding to a specific protein in the body and changing its activity so that symptoms improve. In order to design a small molecule drug that targets a given protein, one must find molecules that bind tightly and specifically to this protein while maintaining properties such as synthesizability, solubility, and permeability. Knowing the 3D structure of the target protein is useful: it reveals the shape of the binding pocket, which in principle allows us to design molecules that fit in and interact with the pocket. Yet even for proteins whose structures have been known for decades, finding molecules that meet all these requirements remains difficult.

A common approach is to first find small chemical groups, known as "fragments," that interact favorably with various parts of a target protein binding pocket—we refer to this step as "fragment generation." Multiple solutions exist for fragment generation, including experimental screening, intuitive design by medicinal chemists, and generative AI techniques (Shim & MacKerell Jr, 2011; Sheng & Zhang, 2013; Lamoree & Hubbard, 2017; Carloni et al., 2025; Powers et al., 2023; 2025; Neeser et al., 2025).

Given a set of fragments and an atom-level representation of a binding pocket, the subsequent challenge is to assemble the fragments into a larger molecule by adding chemical (covalent) bonds. One can predict the inter-fragment covalent bonds between fragments, and then use them to connect the fragments into a complete molecule.

We refer to this task as **molecule assembly**. Broadly speaking, the overarching goal of molecule assembly is to use chemical functional groups or fragments to build a high affinity, druglike ligand for a specific target protein receptor. A druglike ligand is broadly defined as being synthesizable, permeable, metabolically stable, and nontoxic (among many other properties). Molecule assembly is difficult in practice because the fragment positions are known only approximately (or not at all), yielding a large combinatorial space of potential connectivities, each with vastly different pharmacological properties and binding affinities. In machine learning terms, molecule assembly is a traditional graph completion problem, where the nodes are atoms, and the edges are bonds. More specifically, molecule assembly is an example of *spatial graph completion*, where one must predict links between nodes in a spatial graph, where nodes have coordinate noise.

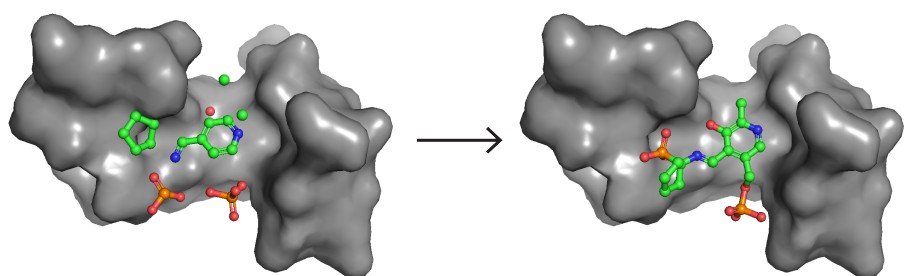

Figure 1: Molecule Assembly Problem Definition. Given fragments (small chemical groups) scattered in a protein binding pocket (left), we wish to predict inter-fragment chemical bonds that will connect them into larger molecule that binds tightly to the pocket (right). The fragments positions and orientations in the larger molecule differ from those provided initially.

No methods directly address spatial graph completion, so we turned to two related tasks: traditional graph completion and all-atom molecule generation. In traditional graph completion, one predicts edges between nodes in a knowledge graph, commonly used in recommender systems and social network analysis (Zamini et al., 2022; Li et al., 2023; Mao et al., 2023). In all-atom molecule generation, one generates all positions, atom types, and bond types of a molecule from scratch, with the goal of designing a strong binder to the target protein.

We adapt methods used for the two aforementioned tasks for molecule assembly. For knowledge graph completion, methods such as GCN and GraphAttention predict edges with the assumption that spatial coordinates do not change (we refer to these as "Traditional graph completion" methods). For all-atom molecule generation, methods such as EDM and Equiformer utilize spatial information to generate atoms and bonds in 3D space (we refer to these as "Geometric deep learning" methods). A full description of these tasks and methods can be found in the Related Work section.

In this manuscript, we find that these methods are ill-suited to the task of molecule assembly. Specifically, traditional graph completion methods do not handle spatial information, and gemoetry-prediction methods do not handle the different classes of edges well—molecule assembly requires learning across fragments, atoms, unknown inter-fragment edges, and known intra-fragment edges.

We thus developed EdGr, a spatial graph diffusion framework to address the molecule assembly problem. EdGr explicitly handles atoms and fragments together in the diffusion pipeline, and ensures all atoms within a fragment are moved according to the same roto-translations. In addition, a dual-edge representation handles intra-fragment edges and inter-fragment edges separately before aggregating and updating coordinate positions. We empirically show that this coupled, dual-edge approach to molecule assembly outperforms classical knowledge graph edge prediction methods (Li et al., 2023; Mao et al., 2023) and recent spatial graph methods (Liao & Smidt, 2022; Hoogeboom et al., 2022). In addition, we compare EdGr to the mixed continuous-discrete all-atom and all-bond diffusion framework used by many ML-based molecule generation methods such as MiDi (Vignac et al., 2023), DecompDiff (Guan et al., 2024), GCDM (Morehead & Cheng, 2024), and DrugFlow (Schneuing et al., 2025), and find that our coupled candidate edge weights-to-coordinates update approach outperforms this framework. In addition, we show that EdGr stays robust despite high amounts of noise in fragment placement.

## 2 RELATED WORK

### 2.1 DIFFERENCES FROM ALL-ATOM MOLECULE GENERATION AND FRAGMENT LINKING

We highlight differences between molecule assembly and two equally important drug design tasks that have previously attracted more attention in the machine learning community: all-atom molecule generation and fragment linking.

**All-atom molecule generation** All-atom molecule generation is defined as follows: given a protein pocket, generate a high-affinity ligand, atom by atom. A variety of diffusion and flow-based models address this task by sampling atom positions, atom identities (element types) (Schneuing et al., 2024; Guan et al., 2023), and bond types (Guan et al., 2024; Morehead & Cheng, 2024; Schneuing et al., 2025; Vignac et al., 2023; Dunn & Koes, 2024). Whereas all-atom molecule generation focuses on creating molecules from scratch, molecule assembly focuses on connecting a known set of approximately placed fragments in a binding pocket. Moreover, all-atom generation does not handle fragments and treats each atom individually.

**Fragment Linking** The fragment linking problem is defined as follows: given two fragments positioned precisely in a protein pocket, create a chain of atoms to link the fragments together. Multiple computational approaches have been developed for fragment linking, including database search (Sheng & Zhang, 2013), autoregressive modeling (Imrie et al., 2020), variational autoencoders (Huang et al., 2022), and diffusion models (Igashov et al., 2024). Fragment linking involves adding linking atoms between immovable fragments, whereas molecule assembly involves adding bonds between noisily placed fragments. This means that fragment linking methods are not applicable to molecule assembly, as each problem requires generating different modalities (bonds for molecule assembly, and atoms for fragment linking).

## 2.2 Existing Spatial Graph Completion Approaches

Two classes of previous developed methods have been applied to knowledge graph completion and all-atom generation: traditional graph completion methods and geometric deep learning methods.

**Traditional graph completion methods** These methods predict missing edges in a graph, treating spatial coordinates as fixed. These methods can be further subdivided into two subclasses: heuristic methods and ML-based link prediction methods.

Heuristic methods predict edges without any learnable parameters, simply relying on properties of the graph to make predictions. For example, the Common Neighbors (Newman, 2001) heuristic computes the similarity of pairs of nodes and links nodes with the highest similarities, and the Minimum Distance heuristic connects nodes that are the closest in physical space.

ML-based link prediction methods (Kipf & Welling, 2016; Veličković et al., 2017) are commonly used for graph completion in the context of knowledge graphs (Zamini et al., 2022; Chaudhri et al., 2021). These methods typically use graph neural networks to learn to impute missing edges in incomplete graphs.

Both subclasses of methods predict new edges in a graph, which fits the bill for molecule assembly. However, graphs in traditional graph completion do not have spatial information. As a result, these methods are not well-suited to molecule assembly, where nodes exist in 3D space.

**Geometric deep learning methods** These methods explicitly predict spatial coordinates of nodes; edges can then be inferred based on methods such as Minimum Distance. Geometric deep learning methods such as EDM (Hoogeboom et al., 2022) and Equiformer (Liao & Smidt, 2022) are explicitly designed for tasks in n-dimensional space, and are popular for molecular applications. The goal of these methods is to predict point positions and attributes, and they are able to do so by treating points in space as nodes in a graph, with edges inferred via a distance cutoff.

Recent work on molecular design has included the development of denoising diffusion and flow models for generating coordinates and edges simultaneously (Dunn & Koes, 2024; Morehead & Cheng, 2024; Guan et al., 2024; Schneuing et al., 2025). Although these methods generate coordinates, elements, and edges simultaneously, they do so via separate diffusion branches, leading to a weak coupling between the atom positions and bond predictions. In contrast, our method's direct coupling approach where bond logits feed directly into the updated coordinates outperforms standard mixed diffusion on molecules (see Results and Tables 1, 2, 3).

## 3 METHODS

### 3.1 DATASET AND SETUP

We follow the dataset preparation steps outlined in Powers et al. (2023) and Powers et al. (2025). Our dataset comprises approximately 35,000 protein-ligand complexes from the Protein Data Bank (PDB). When benchmarking methods on molecule assembly, we measure their ability to reconstruct PDB ligands perfectly, as these ligands are known to be druglike and strong binders. We choose this metric instead of calculating *in silico* druglike metrics because metrics such as QED (Bickerton et al., 2012) have been shown to be unreliable estimators of the aforementioned properties (Beker et al., 2020; Lee et al., 2022; Cai et al., 2022; Li et al., 2024).

We filter out lipids, peptides, nucleic acids, and carbohydrates, and small molecules that are not considered drug-like. The resulting dataset consists of experimentally determined structures of proteins bound to high-affinity, synthesizable, drug-like small molecules. We then split this dataset into train, validation, and test sets (70/15/15), ensuring that the proteins in any given set have less than 30% sequence similarity to any protein in the other sets.

To define the molecule assembly task on this dataset, we take each ligand (i.e., each small molecule) and decompose it into fragments (removing covalent bonds that connect the fragments), following the procedure and fragment library described in Powers et al. (2023) and Powers et al. (2025). This fragment library contains fragments such that double, triple, and aromatic bonds always occur within a fragment rather than between fragments. Our library comprises fragments that are small enough to be treated as inflexible, such as phenyl, methyl, ethyl groups, and benzene rings. Some fragments in our dataset include only one non-hydrogen atom, whereas others include an entire aromatic ring system. Using small fragments allows biologists a greater level of control and precision in fragment selection, as opposed to using large fragments with some sub-groups that are undesirable. The aforementioned bond types are rigid (they cannot be rotated around), so this is consistent with using inflexible fragments. However, one could easily extend this framework to predict double and triple bonds as well. In addition, using small, inflexible fragments makes molecule assembly much more difficult than using large fragments: small fragments mean more fragments in a pocket, yielding a much larger combinatorial search space for bonds compared to using fewer, larger fragments.

### 3.2 FORWARD NOISING PROCESS

We note some key differences between EdGr's forward noising process and that used in standard spatial graph diffusion model. Unlike the standard case, where every point is noised following a closed form multivariate Gaussian distribution, our model treats fragments as inflexible; every atom in a fragment is noised according to the same translation and rotation vector. We add noise to the fragment positions as this reflects the uncertainty in fragment placements outputted by fragment generation methods.

Fragment translational noise is sampled the same way as a standard spatial graph diffusion model samples atom coordinate noise:

$$q(z_t|x) = \mathrm{N}(z_t|\bar{\alpha}_t x, \sigma_t^2 I) \tag{1}$$

$\alpha \in \mathbb{R}^+$ controls the amount of signal retained in original coordinates $x$ and $\sigma^2 \in \mathbb{R}^+$ controls the variance of the normal distribution, in Ångstroms. We do not add noise to the atomic features $h$ as the fragment and atom identities are fixed in molecule assembly. For noising a fragment's orientation, we follow the isotropic Gaussian distribution on SO(3) $g \sim \mathcal{IG}_{SO(3)}(\mu = 0, \epsilon^2)$ (Leach et al., 2022; Savjolova, 1985), which has the density function:

$$f(\omega) = \frac{1 - \cos \omega}{\pi} \sum_{l=0}^{\infty} (2l + 1) e^{-l(l+1)\epsilon^2} \frac{\sin((l + \frac{1}{2})\omega)}{\sin(\frac{\omega}{2})} \tag{2}$$

To noise the bonds, we apply discrete diffusion:

$$q(b_t|b_0) = \text{Bernoulli}(\bar{\alpha}_t b_0 + \frac{(1-\bar{\alpha}_t)}{2}) \tag{3}$$

## 3.3 MODEL AND TRAINING DETAILS

**Preliminaries** We define the following: $h^l$ are node embeddings at layer $l$; $x^l$ are node coordinate embeddings at layer $l$; $s$ is an indicator variable representing self conditioning; $a$ are predefined edge features; $\phi$ are neural networks; $m$ are known edge embeddings; and $n$ are missing edge embeddings. We include pocket atoms in our graph representation, but treat these atoms as static.

**EdGr Implementation** A diagram of the EdGr model can be seen in Figure 2. We have two parallel multi-layer perceptrons (MLPs) to learn edge features, one for known edges and one for missing edges. The known and missing edge features then get aggregated per node and get passed to node MLPs that update node embeddings and positions.

$$s = \begin{cases} 1 \text{ if } U(0,1) < p \\ 0 \text{ otherwise} \end{cases} \tag{4}$$

$$m_{ij} = \phi_e(h_i^l, h_j^l, ||x_i^l - x_j^l||^2, a_{ij}) \tag{5}$$

$$n_{ij;t} = \phi_f(h_i^l, h_j^l, ||x_i^l - x_j^l||^2, a_{ij}, s * n_{ij;t-1}) \tag{6}$$

$$x_i^{l+1} = x_i^l + \frac{1}{M-1}\sum_{j \neq i}(x_i^l - x_j^l)(\phi_x(m_{ij}) + \phi_y(n_{ij})) \tag{7}$$

$$m_i = \sum_{j \in \mathcal{N}(i)} m_{ij} \tag{8}$$

$$n_i = \sum_{j \in \mathcal{N}(i)} n_{ij} \tag{9}$$

$$h_i^{l+1} = \phi_h(h_i^l, m_i, n_i) \tag{10}$$

Equations 5 and 8 are the message passing and aggregation over known intra-fragment edges (Satorras et al., 2022). We add additional candidate inter-fragment edge features $n_{ij}$, which are updated in a similar fashion with a different neural network $\phi_f$ and receive the previous timestep's missing edge embeddings if self conditioning is applied (Equations 4 and 6). Node positions are updated using a sum over all relative distances $(x_i^l - x_j^l)$ (Satorras et al., 2022) multiplied by the sum of the outputs of $\phi_x$ and $\phi_y$, which take in the known edge embeddings $m$ and missing edge embeddings $n$, respectively, and output scalar values (Equation 7). Both edge features are then aggregated across all neighbors of each node $\mathcal{N}(i)$ (Equations 8 and 9) and passed to a node MLP that updates node features (Equation 10). We compute an MSE loss on $x$ and a Binary Cross Entropy Loss on $n_{ij}$.

As shown in the above equations, candidate edge embeddings influence atom positions (Equation 7), and atom positions affect candidate edge embeddings at the following denoising step (Equation 6). This applies to both ligand and pocket atoms — even though pocket atoms are treated as static, the conditioning on the protein pocket guides the movement of atoms and the bond generation possibilities.

## 3.4 INFERENCE

During inference, we ensure that fragments stay rigid during each step of denoising using the Kabsch algorithm (Equation 11) (Lawrence et al., 2019) to calculate the optimal rigid body transformation:

$$\min_{T_t^*, R_t^*} \mathcal{L}(T_t, R_t) = \frac{1}{2}\sum_{i \in \mathcal{F}}||\hat{x}_{i,t+1} - R(\hat{x}_{i,t} + T)||^2 \tag{11}$$

$$x_{t+1} = T_t^* + R_t^*\hat{x}_t \tag{12}$$

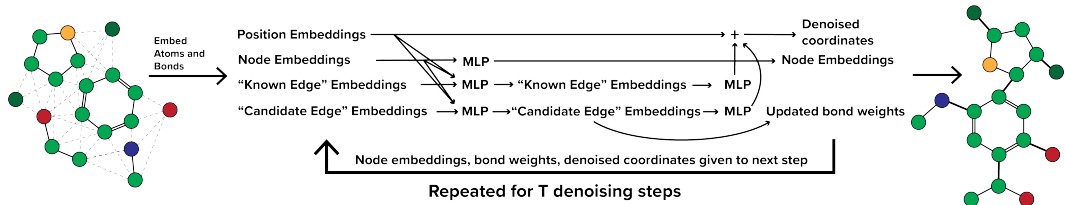

Figure 2: EdGr Architecture Schematic. Node information and edge information are learned through MLPs, and the model outputs updated positions and edge weights (middle). After repeating for T denoising steps, the full molecule with final positions and connectivity is produced (right).

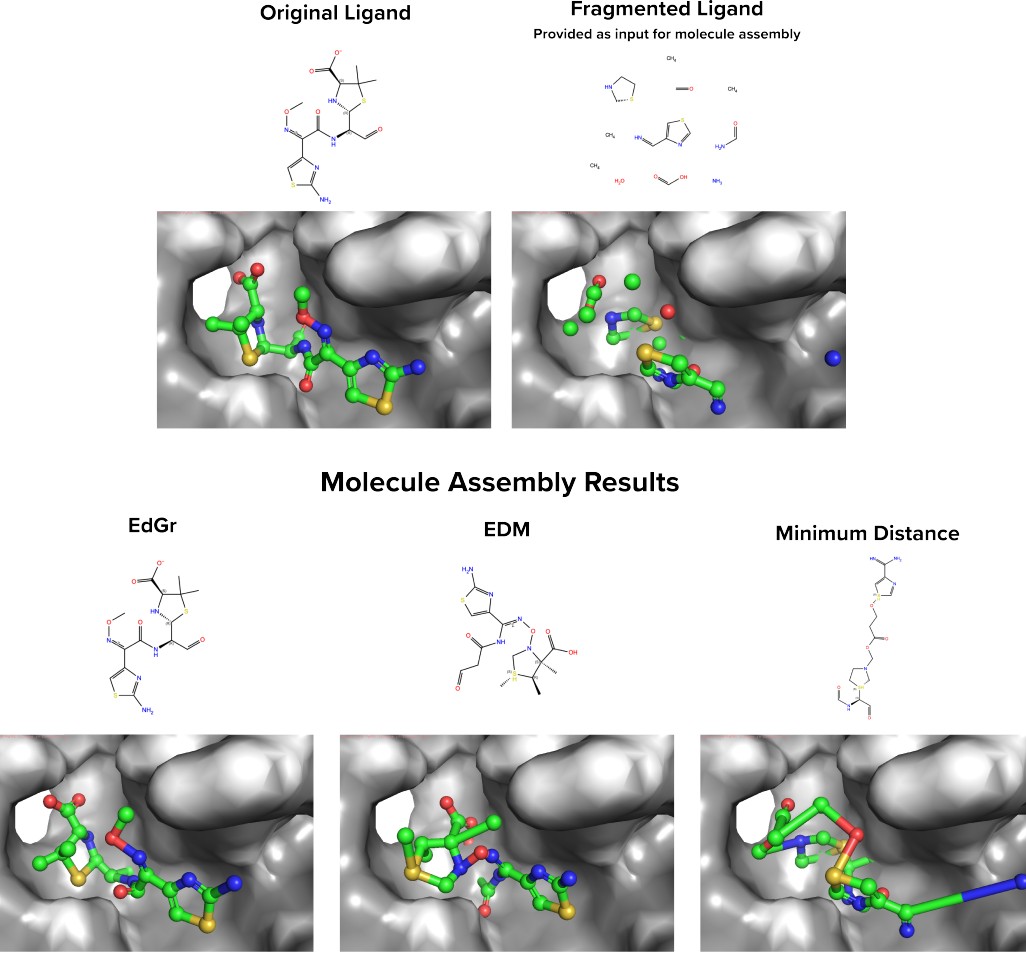

Figure 3: Examples of molecule assembly results from different methods, showing both a 2D graph depiction and a 3D rendering of each molecule. From top to bottom, left to right: the original ligand (a modified version of penicillin, from PDB entry 1LLB); the ligand decomposed into fragments, with rotational and translational noise added to each fragment; molecule assembly results from EdGr, EDM, and the Minimum Distance heuristic.

$\hat{x}_t$ is the predicted locations of atoms in a fragment $\mathcal{F}$ at timestep $t$. $R^*$ and $T^*$ are the optimal rotational and translation vectors, respectively.

After denoising, we obtain our final atom positions and weights for every potential inter-fragment bond within $N$ Ångstroms ($N$ is a cutoff specified as a hyperparameter). To obtain our final list of bonds, we sequentially pop off the bond with the highest weight, check if the bond is chemically plausible and the associated fragments are not connected, and connect the atoms. Full details can be found in Algorithm 1 in the Appendix.

## 4 RESULTS

### 4.1 METRICS

To evaluate model performance, we report the following five metrics. The first four metrics quantify a model's effectiveness at predicting bonds between fragments—the main goal of molecule assembly. The fifth metric quantifies the extent to which atom positions generated by a model match those in the experimentally determined structure.

**Precision & Recall** We define precision and recall as follows:

$$\text{Precision} = \frac{[\text{Predicted Bonds}] \cap [\text{True Bonds}]}{\text{Predicted Bonds}} \tag{13}$$

$$\text{Recall} = \frac{[\text{Predicted Bonds}] \cap [\text{True Bonds}]}{\text{True Bonds}} \tag{14}$$

**Full Molecule Recovery (FMR)** We define "Full Molecule Recovery" as a binary value for each molecule: 0 if the recall is less than 1, and 1 otherwise.

**Tanimoto Similarity** We calculate the Tanimoto coefficient of our recapitulated molecule and the true molecule by first constructing a Morgan fingerprint (Rogers & Hahn, 2010) of both the predicted and original molecule. We use RDKit (Landrum, 2013) to generate the Morgan fingerprint, and use RDKit's builtin Tanimoto Similarity function to calculate the Tanimoto coefficient.

**Root Mean Square Deviation (RMSD)** We also calculate Root Mean Square Deviation (RMSD)—the L2 error between the predicted atom positions and atom positions in the experimentally determined structure (Equation 15). For the molecule assembly task, predicted atom positions are much less important than predicted bonds, but we include this metric because the results may still be instructive.

$$\text{RMSD}(w, v) = \sqrt{\frac{1}{n} \sum_{i=1}^{n} [(v_{ix} - w_{ix})^2 + (v_{iy} - w_{iy})^2 + (v_{iz} - w_{iz})^2]} \tag{15}$$

### 4.2 EXPERIMENTAL SETUP

We split our comparisons table into three types of methods: EdGr, geometric deep learning methods (EDM (Hoogeboom et al., 2022) and Equiformer (Liao & Smidt, 2022)), and traditional graph completion methods (Graph Convolutions (Kipf & Welling, 2016), Graph Attention (Veličković et al., 2017), Minimum Distance heuristic, and Common Neighbors heuristic (Newman, 2001)). We train the geometric deep learning methods to denoise the 3D coordinates in an attempt to recover original atom positions, and then connect the two closest atoms belonging to distinct fragments. For the traditional graph completion methods models, we pass in the molecular graph, treating relative positions between the atom coordinates as edge features, and run standard edge prediction. We do not report RMSD for traditional graph completion methods methods, as they do not change spatial coordinates.

In addition, we benchmark the mixed atom and bond diffusion architectures used by many molecule generation methods (Vignac et al., 2023; Morehead & Cheng, 2024; Guan et al., 2024; Schneuing

et al., 2025) (we report these results as "Mixed Diffusion"). These methods denoise atom types, coordinates, and bond types simultaneously via separate diffusion branches. We train these models in a similar fashion to the geometric deep learning methods, except instead of simply connecting closest atoms, we directly use the predicted bonds from these methods.

### 4.3 COMPARISONS

To evaluate model robustness, we report model performance on different amounts of noise added to the fragments. We report differing amounts of *translational* noise $\mathcal{N}(\mu, \sigma^2)$, where we test $\sigma = 1$Å (Table 1), $\sigma = 2$Å (Table 2), and $\sigma = 3$Å (Table 3, Appendix). A fragment's *rotational* noise is always sampled uniformly from $SO(3)$, meaning that all rotations are equally likely.

To generate confidence intervals, we generate three samples for each ligand in our test dataset (roughly 1,500 examples). We then perform bootstrap sampling to generate a 95 percent confidence interval.

EdGr outperforms all other models tested according to every metric at every level of translational noise. Crucially, EdGr outperforms Mixed Diffusion, where atoms and bonds are generated simultaneously but with separate, uncoupled MLPs, proving the efficacy of the coupled bond-atom framework. After Mixed Diffusion is EDM, which makes uses diffusion to iteratively refine atom positions over $N$ diffusion timesteps. Equiformer attempts to predict the final denoised position in a one-shot fashion, and does poorly. The traditional graph completion methods performed poorly across the board, likely due to their inability to refine coordinate positions, leading to incorrect bond predictions. We see this trend as the noise levels increase — GCN and Minimum Distance, the best-performing traditional graph completion methods, deteriorate in performance. However, EdGr exhibits robust performance despite the increasing amount of noise, with only a 14% drop in precision and a 0.87Å increase in RMSD as the translational noise level increases from $\sigma = 1$Å to $\sigma = 3$Å. We further justify EdGr's design choices through ablation studies, which can be found in the Appendix.

Overall, these results justify the need to develop an architecture specifically tailored to molecule assembly—adapting methods from related tasks underperform. Geometric deep learning methods for all-atom molecule generation either do not predict bonds at all or cannot handle both inter- and intra-fragment edges, and traditional graph completion methods cannot handle noisy coordinate information. Therefore, a framework for molecule assembly that can handle fragments, atoms, and different edge modalities is necessary for addressing this task.

Table 1: Comparison of EdGr to other molecule assembly methods, with translational noise of 1Å standard deviation. Here and in the subsequent tables below, rotational noise is distributed uniformly on $SO(3)$, and error bars show 95 percent confidence intervals determined using bootstrapping. RMSD values are not listed for traditional graph completion methods because those methods do not adjust atom positions.

| Model | Topology | | | | Geometry |
|---|---|---|---|---|---|
| | Precision ↑ | Recall ↑ | FMR ↑ | Tanimoto ↑ | RMSD ↓ |
| EdGr | **85 ± 1%** | **86 ± 1%** | **64 ± 2%** | **88 ± 1%** | **1.09 ± 0.02Å** |
| EDM | 70 ± 1% | 70 ± 1% | 38 ± 2% | 71 ± 1% | 1.20 ± 0.02Å |
| Mixed Diffusion | 79 ± 1% | 79 ± 1% | 53 ± 1% | 83 ± 1% | 1.24 ± 0.02Å |
| Equiformer | 10 ± 1% | 11 ± 1% | 1 ± 0% | 22 ± 0% | 4.46 ± 0.03Å |
| GCN | 23 ± 1% | 21 ± 1% | 2 ± 0% | 29 ± 0% | — |
| Graph Attention | 7 ± 1% | 7 ± 1% | 1 ± 0% | 21 ± 0% | — |
| Minimum Distance | 27 ± 1% | 27 ± 1% | 2 ± 1% | 31 ± 0% | — |
| Common Neighbors | 10 ± 1% | 10 ± 1% | 1 ± 0% | 22 ± 0% | — |

## 5 CONCLUSION

We present EdGr, a graph diffusion-based edge prediction method for molecule assembly that couples prediction of additional bonds with adjustment of atom positions. EdGr substantially and consistently outperforms previous methods for this task, which is important in drug design.

Table 2: Comparison of EdGr to to other molecule assembly methods, with translational noise of 2Å standard deviation.

| Model | Topology | | | | Geometry |
|---|---|---|---|---|---|
| | Precision ↑ | Recall ↑ | FMR ↑ | Tanimoto ↑ | RMSD ↓ |
| EdGr | **76 ± 1%** | **77 ± 1%** | **44 ± 2%** | **74 ± 1%** | **1.58 ± 0.02Å** |
| EDM | 59 ± 1% | 60 ± 1% | 24 ± 1% | 61 ± 1% | 1.65 ± 0.03Å |
| Mixed Diffusion | 69 ± 1% | 69 ± 1% | 34 ± 2% | 70 ± 1% | 1.72 ± 0.02Å |
| Equiformer | 10 ± 1% | 10 ± 1% | 1 ± 0% | 22 ± 0% | 5.25 ± 0.04Å |
| GCN | 11 ± 1% | 11 ± 1% | 1 ± 0% | 22 ± 0% | — |
| Graph Attention | 7 ± 1% | 7 ± 0% | 1 ± 0% | 21 ± 0% | — |
| Minimum Distance | 19 ± 1% | 19 ± 1% | 1 ± 0% | 26 ± 0% | — |
| Common Neighbors | 8 ± 1% | 8 ± 1% | 1 ± 0% | 21 ± 0% | — |

The innovations underlying EdGr—explicit supervision of edge likelihoods and coupled diffusion over coordinates and connectivity—offer a general framework for spatial graph completion. This framework is applicable, in principle, to any graph completion task in which nodes have spatial coordinates that influence edge likelihood and in which knowledge of missing edges would help determine spatial coordinates. For example, in neural connectomics, one wishes to infer fully connected neural circuits from microscopy data in which many connections between neurons are not visible and precise geometries of neurons are uncertain (Ding et al., 2025; Marc et al., 2013). In the computer vision problem of 3D scene reconstruction, one wishes both to determine relationships between objects and to correct for spatial misalignments between objects in images from different view angles (Koch et al., 2024). When designing a wireless sensor network, one must determine both spatial positions of sensors and connectivity between sensors (Khojasteh et al., 2022; Dogan & Brown, 2017). Our results may thus have implications well beyond molecular design.

## REPRODUCIBILITY STATEMENT

Reproducibility details can be found in the Methods section. We will release code for the model in the camera-ready version of the paper.

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

# A  APPENDIX

Table 3: Comparison of EdGr to to other molecule assembly methods, with translational noise of 3Å standard deviation.

| Model | Topology | | | | Geometry |
| --- | --- | --- | --- | --- | --- |
| | Precision ↑ | Recall ↑ | FMR ↑ | Tanimoto ↑ | RMSD ↓ |
| EdGr | **71 ± 1%** | **70 ± 1%** | **34 ± 2%** | **72 ± 1%** | **1.96 ± 0.03Å** |
| EDM | 52 ± 1% | 52 ± 2% | 15 ± 1% | 51 ± 1% | 2.00 ± 0.03Å |
| Equiformer | 10 ± 1% | 11 ± 1% | 1 ± 0% | 22 ± 0% | 6.43 ± 0.04Å |
| GCN | 10 ± 1% | 10 ± 1% | 1 ± 0% | 21 ± 0% | — |
| Graph Attention | 8 ± 1% | 7 ± 1% | 1 ± 0% | 23 ± 0% | — |
| Minimum Distance | 15 ± 1% | 15 ± 1% | 1 ± 0% | 24 ± 0% | — |
| Common Neighbors | 8 ± 1% | 8 ± 1% | 1 ± 0% | 21 ± 0% | — |

## A.1  BROADER SOCIETAL IMPACTS

We believe that this work is important to the field of structure-based drug design, which is highly useful for creating novel drugs to treat diseases and improve human health. However, any such method could also potentially be used to create drugs that do harm. Care is necessary to ensure that this method is used for beneficial purposes.

## A.2  ADDITIONAL BACKGROUND

**Diffusion Models**  Denoising Diffusion Probabilistic Models (Diffusion Models, or DDPMs) (Sohl-Dickstein et al., 2015; Ho et al., 2020), are generative machine learning models inspired by non-equilibrium thermodynamics. They are characterized by two processes: a *forward noising* process which gradually adds Gaussian noise to the original data $x$ via a Markov chain; and a *denoising* process which is parametrized by a neural network $\phi$ that learns to remove the noise.

**Self Conditioning**  In diffusion, $\phi$ learns to either remove the noise $\epsilon$ or directly predict $x_0$ in the chain of denoising steps. However, any intermediate predictions $\tilde{x}_0$ are discarded in the subsequent diffusion steps; self conditioning addresses this deficiency (Chen et al., 2023). Instead of ignoring these intermediate predictions, self conditioning takes these predictions and concatenates them to the noise at timestep $t$ to provide additional context for the model, yielding much better downstream performance (Chen et al., 2023). To prevent the model from becoming too reliant on the intermediate $\tilde{x}$s, we introduce stochasticity with a random variable $s \sim U(0, 1)$; if $s$ is greater than or equal to a preset threshold $p$, self conditioning is not applied.

## A.3  FINAL BOND SELECTION ALGORITHM

## A.4  TRAINING & REPRODUCIBILITY DETAILS FOR EDGR

We train our models on a single Nvidia GPU for up to 300 epochs (approximately 1 week on an Nvidida A40), using the checkpoint with the lowest validation loss for benchmarking. We train all our

---

**Algorithm 1** Final Bond Selection

---

**Require:** List of bonds and model weights for each, ordered from lowest to highest: `bonds`
 1: Initialize QuickFind datatype: `q(n=num_fragments)`
 2: **while** `!q.is_fully_connected()` **do**
 3:    `atom1, atom2 = bonds.pop()`
 4:    `f1, f2 = get_fragment(atom1), get_fragment(atom2)`
 5:    **if** `atom1` and `atom2` are bonded to hydrogen atoms and `!q.is_connected(f1,f2)`
    **then**
 6:        Add `atom1` and `atom2` to final list of bonds
 7:        `q.connect(f1,f2)`
 8:    **else**
 9:        `continue`
10:    **end if**
11: **end while**
12: **return** final list of bonds

---

diffusion models with the AdamW Optimizer, with a learning rate of $3 \times 10^{-4}$, with 100 diffusion steps, batch normalization, using ReLU activations, with 4 hidden layers, each comprising 128 neurons. EdGr receives atom coordinates, element types encoded as one-hot vectors, and fragment membership encoded as a binary vector as input features.

## A.5 TRAINING & REPRODUCIBILITY DETAILS FOR GEOMETRIC DEEP LEARNING METHODS

EDM and Equiformer receive the same input features as EdGr.

**EDM**   We use the EDM architecture from the DiffLinker (Igashov et al., 2024) codebase. We use the same hyperparameters as EdGr ($3 \times 10^{-4}$ learning rate, 4 EGCL layers, each comprising 128 neurons, AdamW optimizer, ReLU activations, 100 diffusion steps, batch normalization). We treat pocket atoms as static and all ligand atoms as flexible. We trained EDM for 300 epochs, saving the checkpoint with the lowest validation loss and using that for benchmarking. To generate the final inter-fragment bonds, we use Algorithm 1, but instead of using the `bonds` list ordered by model weight, the `bonds` list is ordered by Euclidean distance.

**Equiformer**   We use the implementation of Equiformer (Liao & Smidt, 2022) at this GitHub repository: https://github.com/lucidrains/equiformer-pytorch. Due to compute constraints—training an Equiformer model on a single A100 GPU took over a week, with 1 epoch completing every 2 hours—we could not train Equiformer for the full 300 epochs and instead trained it for a week on an A100 (roughly 80 epochs). We saved the model with the lowest validation loss and used that checkpoint for benchmarking. We used the default hyperparameters from the repository, but modified the following: `num_edge_tokens = 2`, `edge_dim = 4`, `single_headed_kv = True`, `heads = 4`, `dim_head = 8`. We generate inter-fragment bonds in the same manner as described in the paragraph describing running EDM.

## A.6 TRAINING & REPRODUCIBILITY DETAILS FOR TRADITIONAL GRAPH COMPLETION METHODS

As additional baselines, we also tested standard implementations of the Graph Convolution Network (GCN) (Kipf & Welling, 2016) and Graph Attention Networks (Veličković et al., 2017) from PyTorch Geometric Version 2.7 (https://pytorch-geometric.readthedocs.io/en/latest/index.html). For both architectures we create a node embedding model that learned node embeddings based on the molecular graphs of all of the provided fragments and a separate link prediction network that took pairs of node embeddings and predicted whether they formed an inter-fragment bond. Each model had nodes that represented atoms, with node features including a one-hot representation of element type, a numerical representation of the fragment identity, the atom's current valence, the maximum number of bonds the atom could form, and a binary flag of whether the atom could form any additional bonds. Learning rates of $1 \times 10^{-2}$, $1 \times 10^{-3}$, and $1 \times 10^{-4}$ were tested for both models. Both methods were trained for 3 epochs on a single GPU with the default settings for the AdamW optimizer

(https://arxiv.org/abs/1711.05101), with the model checkpoints at the end of each epoch featuring the lowest validation loss across all hyperparameters being used to report metrics. During development, additional hyperparameter settings beyond those listed below and longer training times including up to 20 epochs were tested, but did not result in significant changes in validation loss or validation performance. The results reported correspond to the best results obtained from the combination of hyperparameters investigated for these models.

**Running GCN**    The GCN node embedding network featured 3 GCN layers, with a hidden dimension size of 128, and a node embedding output dimension of size 64. The inverted pairwise distances between all atoms based on the noised 3D coordinates were used as edge weights for message passing in the GCN. The ReLU function was used as the non-linear activation function, and dropout layers were placed after the ReLU activation for the first two GCN layers with a dropout rate of 10%. The link prediction network featured 3 MLP layers, with ReLU as the activation function and dropout layers after the first two MLP layers with a dropout rate of 10%. The input dimension for the link prediction network was 128, equal to a pair of node embeddings concatenated together, and the hidden dimension was size 64. The output dimension was size 1, for the binary classification task of whether a given pair of node embeddings should have an inter-fragment bond.

**Running Graph Attention**    The Graph Attention node embedding network featured 3 Graph Attention layers, with a hidden dimension size of 128, a node embedding output dimension of size 64, and 4 attention heads. The pairwise distances between all atoms based on the noised 3D coordinates were provided as edge attributes to each node, along with a one-hot encoding representing whether a given edge was a known intra-fragment bond or a potential inter-fragment bond. The ReLU function was used as the non-linear activation function, and dropout layers were placed after the ReLU activation for the first two GCN layers with a dropout rate of 30%. The link prediction network featured 3 MLP layers, with ReLU as the activation function and dropout layers after the first two MLP layers with a dropout rate of 30%. The input dimension for the link prediction network was 128, equal to a pair of node embeddings concatenated together. The output dimension was size 1, for the binary classification task of whether a given pair of node embeddings should have an inter-fragment bond.

## A.7    ABLATION STUDIES

In Table 4, we report ablations. Removing self conditioning yielded a drop in performance. This was expected, as knowing the model's confidence in the predicted bonds at the previous timestep of denoising should yield an improvement in the following denoising timestep's predictions. In addition, we replace the diffusion trunk of the model with a VAE instead and find that performance on Topology tasks is nearly as good as that of diffusion, but the RMSD is much worse.

Finally, we have previously mentioned the importance of direct edge prediction and the usage of these weights to influence node positions. We investigate the importance of the latter in the following ablation study. We continue to output logits $n_{ij}$ for every candidate edge, but remove these terms from the node coordinate and feature updates. To be precise, we remove the $\phi_y(n_{ij})$ from Equation 7 and $n_i$ from Equation 10. We call this ablation "Remove candidate edge→node update." We find that removing this update while maintaining direct edge prediction results in significantly reduced performance, highlighting the importance of using the candidate edge weights $n_{ij}$ in the updates to the node features $h_i^l$ and coordinates $x_i^l$.

Table 4: Ablation study of EdGr, with translational noise of 1Å standard deviation.

| Model | Topology | | | | Geometry |
| --- | --- | --- | --- | --- | --- |
| | Precision ↑ | Recall ↑ | FMR ↑ | Tanimoto ↑ | RMSD ↓ |
| EdGr base model | $85 \pm 1\%$ | $86 \pm 1\%$ | $64 \pm 2\%$ | $88 \pm 1\%$ | $1.09 \pm 0.02$Å |
| Remove candidate edge→node update | $63 \pm 1\%$ | $64 \pm 1\%$ | $26 \pm 1\%$ | $63 \pm 1\%$ | $1.33 \pm 0.02$Å |
| VAE | $80 \pm 1\%$ | $79 \pm 1\%$ | $51 \pm 2\%$ | $85 \pm 1\%$ | $2.29 \pm 0.02$Å |
| No self conditioning | $80 \pm 1\%$ | $79 \pm 1\%$ | $52 \pm 2\%$ | $83 \pm 1\%$ | $1.28 \pm 0.02$Å |

