# OpenReview forum: "Tackling Molecule Assembly with Graph Diffusion"
_ICLR.cc/2026/Conference — ICLR 2026 Conference Withdrawn Submission_

### Official Review · Reviewer_uNXz · 2025-10-27

**Soundness:** 2
**Presentation:** 3
**Contribution:** 2
**Rating:** 2
**Confidence:** 3

**Summary:**

This paper introduces EdGr, a graph diffusion framework designed for the "molecule assembly" task. This task is defined as connecting a given set of pre-defined molecular fragments into a single, larger molecule within a protein's binding pocket. The model, EdGr, couples the prediction of inter-fragment bonds with the adjustment of fragment/atom coordinates, allowing connectivity information to guide spatial refinement and vice-versa. The authors benchmark EdGr against traditional graph completion methods and geometric deep learning models, demonstrating superior performance in reconstructing the original ligands from their constituent fragments across various noise levels.

**Strengths:**

1. The core idea of EdGr, which couples the diffusion processes for coordinates and connectivity, is novel and technically interesting. Allowing the predicted edge likelihoods to directly influence the coordinate updates is an elegant way to integrate the two modalities.
2. The authors have conducted a comprehensive set of experiments, comparing their method against a reasonable set of baselines from both traditional graph completion and geometric deep learning. The author also includes a rigorous ablation study to demonstrate the effectiveness of each contributing module.

**Weaknesses:**

1. Despite the novelty of the problem and the proposed method, the practical utility of this work is limited. The model's input is a complete set of fragments that constitute the final target ligand. In a real-world FBDD scenario, the primary challenges are (a) identifying one or two initial, weakly-binding fragments from a large library (fragment screening) and (b) subsequently growing or linking these hits with novel chemical matter to improve affinity and properties. One never starts with knowledge of all the fragments of the final molecule. Success on the task of this paper provides no evidence that the model can be useful for FBDD.
2. The model treats fragments as rigid bodies, using the Kabsch algorithm (Section 3.4) to enforce this during inference. Though this choice can indeed reduce the complexity of problems (as we don't need to consider the conformational changes of fragments), it is different from commonly used FBDD libraries known to me, and experiments on this library may not provide enough evidence on the utility for FBDD.

**Questions:**

1. The core premise of the paper rests on the "molecule assembly" task. Could you clarify a practical drug discovery scenario where a medicinal chemist or a computational model would have access to all the constituent fragments of a final, yet-to-be-assembled, high-affinity ligand? The current setup seems disconnected from established FBDD workflows.
2. The paper argues that using small, inflexible fragments gives biologists "greater control." Could the authors elaborate on this point?

---

### Official Review · Reviewer_wjLX · 2025-10-28

**Soundness:** 2
**Presentation:** 3
**Contribution:** 2
**Rating:** 2
**Confidence:** 3

**Summary:**

The paper proposes EdGr, a diffusion-based model for the task of molecule assembly, which is not tackled by previous work according to the authors. Given pre-defined fragments as rigid bodies, the goal is to predict inter-fragment bonds and refine fragment positions so as to reconstruct a full ligand. The key novelty is coupling topology (bond existence) and geometry (node positions) in a unified diffusion framework: each denoising step predicts edge‐likelihoods and uses them to inform coordinate updates, and the updated coordinates feed back into edge prediction.

**Strengths:**

- Clear Motivation: Existing works do not tackle molecule assembly in their setting. All-atom generation treats each atom individually (no concept of fragments), and fragment linking (linker design) does not predict bonds.
- Potential for generalization: The authors point out that the method could apply beyond molecules (neuronal circuits, sensor networks, 3D scene graphs). This broadens interest.

**Weaknesses:**

- The method assumes that a suitable set of fragments is given as input. However, in practice, how would these fragments be obtained? In the paper, if I understand correctly, the authors simply decompose existing ligands from the PDB using computational software. This setup effectively assumes that both the bound ligand and its pocket are known, which is unrealistic for real drug discovery scenarios. Therefore, the practical applicability of the proposed task formulation is unclear.

- The core techniques appear to be standard/not novel. Section 3.2 is just regular diffusion formulation, and Section 3.3 essentially implements a variant of EGNN with an additional edge MLP branch. The main contribution lies in framing these components for the “molecule assembly” problem, but the novelty in terms of algorithmic design is limited. Furthermore, since the practicality of the problem setting itself is questionable, the overall impact of the work is diminished.

- The model treats each fragment as a rigid body that can only undergo global translation and rotation. In real molecular systems, fragments can deform or exhibit internal flexibility when binding to a protein pocket. This rigid-body assumption may make the results unrealistic for true molecular design tasks. It would be helpful to discuss whether intra-fragment flexibility could be incorporated or whether the method’s performance degrades if fragments deviate from rigidity.

- he paper does not evaluate or enforce the chemical validity of the assembled molecules. There is no check for valence satisfaction, ring closure correctness, or physically plausible geometries. Without such checks, it is unclear whether the generated assemblies correspond to chemically realizable molecules.

**Questions:**

1. How would EdGr behave if the input fragments came from an actual fragment-screening experiment (rather than decomposition of known ligands)?
2. Since the dataset is derived by decomposing PDB ligands, the model effectively learns to “reconnect” existing fragments. Does this introduce data leakage or trivial reconstruction patterns (e.g., memorizing frequent bond patterns)?

---

### Official Review · Reviewer_41T9 · 2025-10-30

**Soundness:** 2
**Presentation:** 1
**Contribution:** 2
**Rating:** 2
**Confidence:** 3

**Summary:**

The paper frames molecule assembly as a spatial graph completion problem, where fragments with noisy coordinates must be connected into a plausible ligand. It introduces a diffusion-based model, EdGr, that simultaneously updates atomic positions and bond predictions. Fragments are treated as rigid bodies subject to rotations and translations, while candidate bond logits directly influence coordinate denoising. The architecture maintains separate channels for intra-fragment and inter-fragment edges, which are later aggregated for node updates. Inference proceeds by sequentially selecting high-weighted bonds and enforcing rigid fragment alignment

**Strengths:**

- The task tackled in the paper seems relatively novel, though my knowledge of the field for fragment linking is rather limited.

- Though the choice of baselines could be considered questionable (see weaknesses), it seems that the methods performs well on the selected benchmark.

**Weaknesses:**

- l.80: “methods such as EDM and Equiformer” : it would be preferable to provide explicit references here, even if these methods are described later.

- The Methods section begins directly with a dataset description. This belongs in the experimental section, not in the methodological presentation.

- Notations are often undefined, and the description of the diffusion model remains rather vague. Overall, diffusion models are not thoroughly introduced, without proper background or context, and too much is assumed as prior knowledge.

- Using separate neural networks for intra- and inter-fragment edges doubles the memory complexity.

- The choice of baselines is questionable: comparing 3D geometric methods to GNNs designed for non-geometric tasks lacks practical relevance. More appropriate baselines would be fragment-based molecular generation methods (see [1]).


[1] Voloboev, S. (2024). A Review on Fragment-based De Novo 2D Molecule Generation. arXiv preprint arXiv:2405.05293.

**Questions:**

- Why is the Kabsch algorithm needed at inference time if the model design already ensures that atoms within a fragment are treated equivalently?

- A major concern with the experimental setup is the choice of fragments. The authors appear to assume that the correct fragments are already identified and only need to be linked. In practice, however, the real challenge lies precisely in determining which fragments should be combined in the first place.

---

### Official Review · Reviewer_WaUo · 2025-10-30

**Soundness:** 1
**Presentation:** 2
**Contribution:** 1
**Rating:** 2
**Confidence:** 3

**Summary:**

This paper introduces EdGr, a spatial graph diffusion framework designed for assembling molecular fragments. The method sees fragment as rigid, and iteratively updates atom positions and inter-fragment edges.  EdGr consistantly outperforms other methods.

**Strengths:**

1.The paper is clearly written and presents its ideas in a detailed and structured manner.
2.This is an interesting technical problem, although it may ot directly address a real-world need in molecular design.

**Weaknesses:**

1.The task isn’t really necessary or challenging in real drug design. In practice, the focus is on generating molecules or generating and selecting useful fragments. Just connecting fragments that are already given is much less challenging.
2.The paper mentions other applications like neural connectomics, 3D scene reconstruction, and wireless sensor networks, but these are not tested in the experiments and don’t match the protein–ligand dataset used, so the connection feels a bit forced.

**Questions:**

Can the authors clarify why molecule assembly—connecting pre-defined fragments—is chosen as the main task?

---

### Official Review · Reviewer_MVpj · 2025-11-03

**Soundness:** 2
**Presentation:** 2
**Contribution:** 2
**Rating:** 2
**Confidence:** 4

**Summary:**

The paper introduces EdGr, a spatial graph diffusion framework for molecule assembly—connecting approximately placed fragments inside a protein pocket into a complete, drug-like ligand. EdGr jointly handles fragments and atoms, keeps fragments rigid during denoising, and couples candidate inter-fragment bond logits to coordinate updates so connectivity cues steer spatial movements. The method is evaluated on ~35k PDB protein–ligand complexes (train/val/test split with <30% protein sequence identity between splits) by fragmenting ground-truth ligands into small, mostly inflexible fragments; double/triple/aromatic bonds are constrained to remain within fragments. Metrics include Precision/Recall/FMR (exact topology recovery), Tanimoto similarity, and RMSD. Across noise levels (σ=1–3 Å translational noise; uniform rotational noise on SO(3)), EdGr reports higher precision/recall/FMR and lower RMSD than EDM, Equiformer, Mixed Diffusion, and traditional link-prediction heuristics/GNNs.

**Strengths:**

- Treating fragments as rigid (shared roto-translation noise), coupling edge logits into coordinate updates, and enforcing rigidity via Kabsch at each step are clearly specified and align with the problem geometry.
- The dataset construction, split policy, and evaluation criteria (Precision/Recall/FMR, Tanimoto, RMSD) are spelled out, and comparisons span geometric, mixed diffusion, and graph-completion baselines.
- EdGr consistently beats baselines under increasing placement noise and includes ablations (self-conditioning, candidate-edge→node coupling).

**Weaknesses:**

- The receptor is held rigid and noise is isotropic, which sidesteps induced-fit and realistic perturbations. Can you evaluate with receptor flexibility (e.g., side-chain/backbone motion or flexible-receptor docking starts)? How does performance change under empirically grounded, non-isotropic noise models? How sensitive are results to pocket perturbations?
- Enforcing rigid Kabsch moves forbids intra-fragment strain or ring puckering. Can you run ablations that allow limited intra-fragment flexibility (e.g., selected rotors or ring torsions), and report the impact on accuracy and stability?
- Bond selection relies on ad-hoc rules and differs across methods. Can you standardize post-processing across all methods or report “model-only” results without post-processing? Can you quantify how each heuristic changes outcomes (success rate, RMSD, topology)?
- Training budgets and tuning differ markedly across baselines. Will you retrain baselines under matched compute or matched early-stopping criteria? Can you provide hyperparameter-sweep details and multi-seed statistics to ensure parity?
- Metrics emphasize topology/RMSD over binding or developability. Can you add chemically grounded validity checks (valence, clashes, strain) and target-aware proxies (pose recovery, docking/physics scores)? Can you increase sampling to tighten confidence intervals?
- Results hinge on a narrow set of noise scales/distributions. Will you sweep translational/rotational noise magnitudes and non-uniform distributions, and report robustness curves and failure modes?
- Core assumptions (rigid fragments, static receptor, post-hoc bonding) aren’t ablated. Can you ablate or relax each assumption in turn and quantify its contribution to performance?

**Questions:**

See the weeknesses.

**Details Of Ethics Concerns:**

NA.

---

### Note · Authors · 2025-11-13

I have read and agree with the venue's withdrawal policy on behalf of myself and my co-authors.